# Recent Advances in the Use of CoPc-MWCNTs Nanocomposites as Electrochemical Sensing Materials

**DOI:** 10.3390/bios12100850

**Published:** 2022-10-09

**Authors:** Sheriff A. Balogun, Omolola E. Fayemi

**Affiliations:** 1Department of Chemistry, Faculty of Natural and Agricultural Sciences, North-West University (Mafikeng Campus), Mmabatho 2735, South Africa; 2Material Science Innovation and Modelling (MaSIM) Research Focus Area, Faculty of Natural and Agricultural Sciences, North-West University (Mafikeng Campus), Mmabatho 2735, South Africa

**Keywords:** cobalt phthalocyanines, multiwalled carbon nanotubes, electrochemical sensors, electrochemical techniques

## Abstract

Cobalt phthalocyanine multiwalled carbon nanotubes (CoPc-MWCNTs), a nanocomposite, are extraordinary electrochemical sensing materials. This material has attracted growing interest owing to its unique physicochemical properties. Notably, the metal at the center of the metal phthalocyanine structure offers an enhanced redox-active behavior used to design solid electrodes for determining varieties of analytes. This review extensively discusses current developments in CoPc-MWCNTs nanocomposites as potential materials for electrochemical sensors, along with their different fabrication methods, modifying electrodes, and the detected analytes. The advantages of CoPc-MWCNTs nanocomposite as sensing material and its future perspectives are carefully reviewed and discussed.

## 1. Introduction

Developing new electrochemical sensing materials with enhanced electrocatalytic properties, good stability, reproducibility and repeatability, high sensitivity, and selectivity is one of the most significant and rapidly growing areas in materials science [1]. Carbon nanotubes (CNTs) have become promising materials in constructing sensors and biosensors for wider applications simply because of their excellent chemical inertness, high specific surface area, high mechanical strength, and high electrical conductivity with a unique one-dimensional structure that allows rapid electron transfer [2]. In addition, CNTs can be found in a wide range of applications, e.g., in electronics, polymer composites, energy storage materials, catalysis, gas storage materials, and sensors [3]. Particularly, multiwalled carbon nanotubes (MWCNTs), as depicted in Figure 1, exhibit good mechanical strength and enhanced surface activity with a high specific surface area. They are frequently utilized in biological applications, thermally stable materials, sensors, water filtration, structural materials, and so forth [4,5]. Additionally, the acid functionalized MWCNTs (fMWCNTs) are very intriguing as catalyst supports because of their multiple means of connecting with organic molecules [6]. Studies have shown that during redox reactions in both acidic and alkaline mediums, electron transfer occurs rapidly with MWCNTs, resulting in higher current density and lower redox potential. Thus, MWCNTs have better redox activity than single-walled carbon nanotubes (SWCNTs) [7]. Importantly, CNTs (SWCNTs and MWCNTs) have been extensively used as a sensing material to fabricate various nanocomposites which have been successfully employed to determine a wide range of analytes, such as uric acid, ascorbic acid, dopamine [8,9], styrene, epinephrine [7,10], glutathione, cysteine, acetaminophen [11,12], carbaryl, thiols, bisphenols [13,14,15], bromate, nitrite [16], hydrogen peroxide, paracetamol [17,18], lactic acid, hydrazine, glucose [19,20,21], and others.

Phthalocyanines (Pc) are macrocyclic compounds with a central planar molecule and an 18-π-electrons electron system, which is delocalized on the carbon-nitrogen double bond [22]. Phthalocyanines are one of the best chemicals with broad applications in electrocatalysis, electrochemical sensors, electrochromism, photodynamic therapy, photovoltaic cells, photosensitizer liquid crystal materials, and catalysts due to their high stability and good spectral performance [23,24,25]. Pcs can bind to a variety of analytes in an indifferent manner via the coordination interactions with the central metal, hydrogen bonds, and van der Waals forces [26]. They are highly suitable for incorporating into electrochemical sensors due to their significant chemical and thermal stability qualities. Additionally, the chance to incorporate up to 70 different metal atoms into their rings and the ability to vary the side chain substituents result in the formation of unique and efficient thin films with varying degrees of stability, selectivity, and sensitivity [22].

Recently, metal phthalocyanines (MPcs) have received much interest in the field of catalysis. The high catalytic activity of metal phthalocyanine is attributed to the central metal ions [6,27]. Among these MPcs, cobalt phthalocyanine (CoPc) is one of the bright catalysts for varied organic reactions because of its high activity and selectivity of oxidation processes [6,28,29,30]. CoPc has been widely used as a mediator in electronic devices and in constructing electrochemical sensors due to its electronic, catalytic, and semi-conductor proprieties; besides, it can be anchored in cationic substrates by simple adsorption processes [31,32]. Additionally, the rich redox chemistry of CoPc and its high ability to transport electrons leading to its outstanding electrocatalytic activity for various chemicals, has given it a wider usage in sensor fabrication [19].

The coordination environment of the central metal of the CoPc, as shown in Figure 1, can be changed easily which effectively enhances its catalytic activity and selectivity. One of the outstanding properties of CoPc, which makes it an excellent material in electrochemical sensor fabrication, is the ease of replacement of organic groups in the axial and equatorial positions of the complex, giving rise to several functionalities for anchoring complexes in solid substrates [33,34,35,36]. In metal phthalocyanine, the central metal ions have six coordination sites, in addition to four N coordination with the central metal, there are also two coordination sites. The metal phthalocyanine can be fixed to the carrier in an axial coordination manner by introducing an axial ligand, improving the activity, selectivity, and stability of the catalyst [37,38,39,40].

Multiple series of approaches have been developed for the synthesis of CoPcMWCNTs including ultrasonic impregnation, solid phase synthesis, chemical deposition, amide bridge synthesis, electropolymerization, and drop coating method. The formation of CoPcMWCNTs nanocomposites is accomplished by the π-π interactions between the -COOH of acidified functionalized MWCNT and the -NH_2_ terminal of CoPc composites. Through these π-π interactions, CoPc molecules can be anchored on the wall of MWCNTs by axial coordination to achieve molecular dispersion of CoPc on MWCNTs. The axial coordination of metal phthalocyanine also provides another driving force for immobilization, which invariably increases electrocatalytic activity, selectivity, and stability of the nanocomposites. Additionally, the high solubility of CoPc offers a huge advantage in fabricating several sensors due to its film forming capacity with a positively charged polyelectrolyte, which are widely used in the layer-by-layer (LbL) method and LB method of film forming [41,42,43].

Researchers have found that phthalocyanine–CNT complexes exhibit excellent catalytic properties of Pc while retaining all the electronic properties of carbon nanotubes [44]. Moreover, Pcs have been widely studied for functionalizing CNTs because they demonstrate rich electronic and photoelectronic properties, which make CNT-based devices more efficient. Recent reports show that MPc-CNT hybrids exhibit enhanced electrochemical responses compared to CNTs or MPc alone [45]. According to reports in the literature, redox overpotentials are decreased, and electrode faradaic currents are increased when thin films of MPcs are immobilized on working electrodes together with other highly conductive materials [46,47]. Noteworthy, incorporating metallic nanoparticles and carbon-based components (graphene, sheets, MWCNTs, quantum dots, SWCNTs) into the MPc-based film increases its conductivity [48]. Through the use of these conductivity-enhancing materials, fast electron transfer is promoted between the surface of the electrode and analytes adsorbed on the thin film. In addition, electrochemical sensors produced from MPcs and other nanomaterials exhibit large electroactive surface areas for an improved immobilization of analytes compared to bare electrodes [46].

CoPcMWCNTs nanocomposites have recently been studied and exhibit improved capacitive behavior due to the enlarged surface area of the phthalocyanine and MWCNTs as well as the excellent conductivity and stability of cobalt and the MWCNTs [49]. The top qualities of MWCNTs, cobalt (Co), and Pc, are combined to construct electrochemical sensors with a high current response, huge capacitance, remarkable cycling stability, good repeatability and reproducibility, high sensitivity, and selectivity. Electrochemical sensors fabricated with CoPc-MWCNTs nanocomposites have been employed to determine numerous types of analytes with outstanding performances ranging from the low limit of detection, high sensitivity to a wider concentration range. Hence, the main reason for the wide usage of CoPc-MWCNTs nanocomposite in fabricating electrochemical sensors either alone or in combination with other conducting materials. In this review, we have extensively discussed the detection of various analytes using electrochemical sensors fabricated with CoPc-MWCNTs nanocomposite. Details of these are given below.

## 2. Application of CoPc-MWCNTs Nanocomposite as Electrochemical Sensing Material

### 2.1. Detection of Ascorbic Acid, Dopamine, Paracetamol

Kutluay and Aslanoglu [50] developed an electrochemical sensor to simultaneously detect dopamine (DA) and paracetamol (PAR) via chemical deposition. The SEM image of the prepared CoNPs/MWCNT/GCE clearly shows the surface of the MWCNTs having a spherical and evenly dispersed CoNPs. The CoNPsMWCNT-GCE could simultaneously detect DA and PAR in 0.1 M phosphate buffer saline PBS (pH 7) using square wave voltammetry (SWV). The electrode exhibited good linear concentration ranges of 0.05–3.0 μM and 0.0052–0.45 μM for DA and PAR, respectively. Limit of detections (LOD) of 0.015 and 0.001 μM were obtained for DA and PAR, respectively. The developed electrode exhibited good stability, reproducibility, repeatability, and high recovery. An interference study showed that UA and AA did not affect the detection of PAR and DA. This technique accomplished the determination of PAR and DA in pharmaceutical drugs.

Xia Zuo and his research group [51] fabricated CoPc-MWCNTs/GCE electrode via a drop coating method to determine ascorbic acid (AA). The fabrication of CoPc-MWCNTs nanocomposites was actualized by exploiting the advantage of strong non-covalent interactions between the highly delocalized π-bonding network of MWCNTs and the metal phthalocyanines conjugated ring. This consequently influenced the electrode performances having strong electrocatalytic activity toward AA with the oxidation potential of the AA simultaneously decreased. The study revealed that the fabricated electrode exhibited enhanced electrocatalytic behavior towards the oxidation of ascorbic acid in 0.1 M PBS at a pH of 7. The biosensor offered an LOD of 1 μM and a wider concentration range of 1.0 × 10^−^^5^ M to 2.6 × 10^−^^3^ M. The CoPc-MWCNTs/GCE electrode is characterized by high stability, high reproducibility, and fast response time. The mechanism of oxidation of AA at CoPc-MWCNTs/GCE involves two-step electrocatalysis. The first step is the oxidation of Co^II^Pc to Co^III^Pc while the second step is the chemical oxidation of AA and the regeneration of Co^II^Pc. These are illustrated in Figure 1 below.

A series of complex interactions occurs between the central metal ion and the activated AA as it diffuses from the bulk solution to the sensor surface and is adsorbed on the macrocyclic 18-electron conjugated system of phthalocyanine that is bonded to the MWNTs surface. The Co^III^/Co^II^ absorb the electrons lost from AA and transfer them to MWNTs on the hydrophobic surface of the GCE. The authors also established that the CoPc-MWCNTs/GCE modified electrode exhibited good selectivity for AA determination in the presence of glucose, potassium chloride, l-phenylalanine, citric acid, and uric acid.

Jilani et al. [52] fabricated a GCE/MWCNT-CoTMBANAPc electrode to simultaneously detect ascorbic acid and dopamine in PBS at a pH of 7.0 via AP and differential pulse voltammetry (DPV). The Tetra8[(*E*)(4methoxybenzylidene)amino] naphthalene1amine cobalt (II) phthalocyanine (CoTMBANAPc) was synthesized from cobalt (II) tetracarboxylic acid phthalocyanine (CoTCAPc) via amide bridge. The oxidation process of AA and DA at GCE/MWCNT-CoTMBANAPc electrode surface involves a three-step electrocatalytic process. The first two steps are the same as those explained in Figure 1 while the third step, which involves the chemical oxidation of DA and the regeneration of Co^II^Pc, is illustrated in Figure 2 below.

The fabricated sensor was able to determine both AA and DA with a low LOD of 0.33 and 6.6 μM for DA and AA, respectively, and a limit of quantification from 1 to 15 μM.

The authors reported that the fabricated sensor exhibited high selectivity for AA and DA amidst interferents, such as glycine, glucose, hydrogen peroxide, tyrosine, and L-cysteine. The stability, reproducibility, and sensitivity of the fabricated electrode were good.

Another CoPc-MWCNTs-based sensor for dopamine and paracetamol detection from Mounesh and Reddy [53] was produced by drop-casting tetra 1-benzyl-1H-pyrazol-3-carboxamide cobalt(II) phthalocyanine (CoTBPCAPc) with MWCNTs on a GCE’s surface. The electrocatalytic mechanism of DA at CoTBPCAPc/MWCNTs/GCE is illustrated with Equations (1) and (2). CoPc(III) oxidizes dopamine to dehydrodopamine (DHDA) by the center Co ion of the phthalocyanine and then regenerates CoPc(II).
CoPc(II) → CoPc(III) + e^−^ + H^+^(1)
2CoPc(III) + DA → 2CoPc(II) + DHDA + 2H^+^(2)

In 0.1 M PBS (pH 7), the CoTBPCAPc/MWCNTs/GCE electrode’s electrocatalytic activity towards DA and PAR was performed using cyclic voltammetry (CV), DPV, and chronoamperometry (CA). The developed sensor displayed excellent DA and PAR detection performance in linear ranges of 50–750 nM, with a LODs of 17 and 19 nM, respectively. The authors described the sensor as having high stability, excellent reproducibility, and repeatability. When applied to detect DA and PAR in both commercial and urine samples, the sensor delivered satisfactory results.

Again, Moraes and his research team [54] used CoPcMWCNTs/GCE modified electrodes to fabricate an electrochemical sensor to detect dopamine in ascorbic acid. Under optimum conditions, in 0.2 M PBS (pH 4.0), dopamine was successfully detected by the CoPcMWCNTs/GCE modified electrode via DPV with a low LOD of 0.256 μM and a wide LDR of 3.11–93.2 μM. The authors described the sensor as reliable for determining DA due to its exceptional qualities, including good stability, repeatability and reproducibility, high sensitivity, and selectivity.

A summary of the analytes detected with CoPc-MWCNT-modified electrodes with the techniques employed, LOD, and LDR is given in Table 1 below.

### 2.2. Detection of Hydrogen Peroxide, Nitrite, and Heavy Metals

Mounesh and Reddy [56] accomplished the fabrication of CoPcMWCNTs/GCE for the detection of heavy metals (Pb^2+^ and Cd^2+^) in water. Electro-spraying was specifically used to modify the CoTEIndCAPc/MWCNTs on the GCE since it forms particles with a sizable specific surface area that can potentially provide a fast electron transfer and ultra-high sensitivity for Cd^2+^ and Pb^2+^ detection. The modified CoPc-MWCNTs/GCE detected both Pb^2+^ and Cd^2+^ with very low LOD of 9 and 10 nM in PBS at a pH of 7.0 via DPV and CA techniques. The electrocatalytic property of the CoPc-MWCNTs/GCE electrode was characterized by high reproducibility and repeatability. This sensor displayed an excellent selectivity for both Cd^2+^ and Pb^2+^ in the presence of other similar ions (Al^3+^, Ca^2+^, Cu^2+^, Mg^2+^, Mn^2+^, Zn^2+^, As^3+^, Cr^3+^, and Fe^3+^). The sensor’s excellent selectivity for Cd^2+^ and Pb^2+^ ions is attributed to the selective movement of the analyte ions from the buffer solution to the electrode surface owing to the stronger affinity of the CoTEIndCAPc/MWCNTs/GCE electrode for Cd^2+^ and Pb^2+^ ions. The electrode was also applied to determine Pb^2+^ and Cd^2+^ in river water.

Mounesh and Reddy [26] fabricated another CoPc-MWCNTs/GCE to simultaneously detect hydrogen peroxide (H_2_O_2_) and nitrite using CV, CA, and DPV techniques. The best electrocatalytic oxidation of CoPc-MWCNTs/GCE was achieved in PBS solution (pH 7.0) with high stability, repeatability, reproducibility, and sensitivity. The mechanism of NO_2_ and H_2_O_2_ oxidation at the CoTL-MethPc/MWCNTs/GCE are given in Equations (3)–(5).
CoPc(II) → CoPc(III) + e^−^ + H^+^(3)
2CoPc(III) + NO_2_ + H_2_O → 2CoPc(II) + NO_3_ + 2H^+^(4)
2CoPc(III) + H_2_O_2_ → 2CoPc(II) + 2H_2_O + 2H^+^(5)

The authors found that the electrocatalytic oxidation of CoPc-MWCNTs/GCE towards nitrite and hydrogen peroxide was diffusion controlled with specific adsorption of electro-redox process and intermediates products. The electrode offered a good linear concentration between 0.1 and 0.8 μM and low LOD of 10 nM and 30 nM for H_2_O_2_ and nitrite via CV. The developed sensor utilized the advantage of simple preparation, low cost, high selectivity, and real sample application (in beetroot vegetable).

Lu et al. [57] presented a nitrite sensor using a CoPcMWCNTs nanocomposite GCE-modified electrode. Using the DPV technique, the developed CoPcMWCNTs/GCE electrode detected nitrite in 0.1 M PBS at a pH of 7.4 in a linear dynamic range (LDR) from 0.01 to 1050 mM and a detection limit of 2.11 μM. Thus, the electrode is said to be a facile, sensitive, and rapid electrochemical technique for detecting nitrite. Aside from this, it also offers good stability, sensitivity, and reproducibility. The outstanding performance of the fabricated sensor is attributed to the synergistic interaction of MWCNT and CoPc, thereby demonstrating the potential applications of CoPcMWCNTs/GCE in real-life biosensing analysis. The sensor’s selectivity investigation showed that nitrite detection was unaffected by the presence of these interferents (CH_3_COONa, NaCl, NaNO_3_, Na_2_SO_4_, KCl, ascorbic acid, and glucose).

Table 2 below summarizes the analytes detected with CoPc-MWCNT-modified electrodes with the techniques employed, LOD, and LDR.

### 2.3. Detection of Carbaryl, Acetaminophen, Epinephrine, and Procalcitonin

Moraes and his research group [59] fabricated a sensor by modifying GCE with MWCNTs/CoPc nanocomposites, which was prepared through ultrasonic impregnation of CoPc onto the MWCNTs. The fabrication and modification process of CoPcMWCNTs nanocomposite with GCE is shown in Figure 2. The fabricated MWCNTs/CoPc/GCE film electrode was employed to detect carbaryl in acetate buffer solution (pH 4.0) via SWV. The sensor gave an LDR and LOD of 0.33–6.61 µM and 5.46 ± 0.02 nM, respectively. The sensor was proven suitable in carbaryl-spiked water samples.

Aragao and his research group [60] developed an electrochemical sensor for diethylstilbestrol (DES) detection by modifying GCE with gold (Au) nanoparticles and CoPcMWCNTs nanocomposites via electrodeposition. Prior to this, the synthesis of the nanocomposites was achieved through an ultrasonication method. The mechanism of the oxidation of DES molecules on the CoPcMWCNTs/AuNPTs/GCE (which was mainly an adsorption-controlled process) involves two-step processes with one electron each. The first step is the oxidation of neutral DES molecule to phenoxy ion, while the second step is the formation of phenoxonium anion. The developed CoPc-fMWCNTs/Au/GCE catalyzed diethylstilbestrol’s electrochemical oxidation and enhanced its sensitivity compared with the unmodified electrode. Through the SWV technique, DES was determined in Britton–Robinson (BR) buffer solution (pH 10) with an LOD of 0.199 µM and quantification limits of 0.664 µM. An interference study showed that most ions and molecules (K^+^, Na^+^, Ca^2+^, Mg^2+^, Zn^2+^, Pb^2+^, Al^3+^, Cl^−^, NO_3_^−^, SO_4_^−^, H_2_PO_4_^−^, ascorbic acid, citric acid, glucose, uric acid, dopamine, and urea) did not interfere with the detection of DES, indicating the good anti-interference ability of the electrode. Good reproducibility and repeatability exhibited by the developed electrode indicates the suitability of the electrode modification for DES detection. The proposed sensor was successfully employed to determine DES in meat and water with satisfying outcomes, indicating that the developed sensor is reliable for DES detection in complex samples.

To present a sensing platform for acetaminophen detection, Kantize et al. [46] ultrasonically fabricated CoPc/MWCNTs nanocomposites which were then modified on a GCE using a drop-dry method. The CoPcMWCNTs/GCE electrode exhibited good stability and sensitivity in CV experiments. The best electrocatalytic behavior of the electrode for acetaminophen detection was achieved in 0.1 M PBS at a pH of 7.4, giving an LOD of 1 μM and LDR of 0.975–1000 μM. The fabricated electrodes also showed excellent selectivity when tested in various pharmaceutical interferent species.

Furthermore, Holanda and his research group [61] used functionalized multiwalled carbon nanotubes, gold nanoparticles, cobalt (II) phthalocyanine, and GCE for fabricating a sensor for acetaminophen detection. The preparation involves ultrasonic mixing of fMWCNT and CoPc in dimethylformamide (DMF) followed by drop-drying the fMWCNT-CoPc suspension on the AuNPs/GCE to obtain the fMWCNT-CoPc/AuNPs/GCE. A very low charge transfer resistance (Rct) value obtained for the fMWCNT-CoPc/AuNPs/GCE modified electrode via an EIS experiment confirms that the modification of fMWCNT-CoPc and AuNPs onto the GCE significantly enhanced the electron transfer, predominantly due to the high conductivity of AuNPs and the improved catalytic activity from the synergy effect between CoPc and fMWCNTs since the CoPc acts as a charge transfer mediator. The mechanism oxidation of acetaminophen on the fabricated fMWCNT-CoPc/AuNPs/GCE is illustrated in Figure 3 below. The fMWCNT-CoPc/AuNPs/GCE was used to detect acetaminophen in McIlvaine buffer solution (pH 5) by SWV. The fabricated sensor reached a detection limit of 0.135 μM within an LDR from 1.49 to 47.6 μM. To assess the usefulness of the fabricated sensor, acetaminophen was detected in four various sample matrices of commercial pharmaceuticals with good recovery.

Another electrochemical sensor was produced by Moraes and his research team [62] for the detection of epinephrine (EP) in urine using a paraffin composite electrode PCE modified with cobalt phthalocyanine and multiwalled carbon nanotubes. The mechanism of EP oxidation on the CoPcMWCNT/PCE electrode is illustrated in Figure 4 below. In 0.1 M PBS at a pH of 6.0, the CoPcMWCNT/PCE electrode successfully detected epinephrine via DPV with an LOD of 15.6 nM and a wide LDR of 1.33–5.50 µM. The authors established that the sensor is reliable for determining epinephrine due to its good stability, repeatability and reproducibility, sensitivity, and selectivity.

Agboola and his research team [63] constructed another sensing platform for epinephrine detection by modifying edge-plane pyrolytic graphite (PG) electrodes with CoPc and acid-functionalized SWCNTs. The modification process involves the drop-dry method and electrodeposition. SWCNTs were first modified on the PG electrode by the drop-dry method, followed by the electrodeposition of the CoPc complex on the PG containing SWCNT. The mechanism oxidation of EP on the PG/SWCNTs-CoPc electrode as illustrated in Figure 5 below involves the oxidation of epinephrine to epinephrinequinone via removal of the two protons of the enol end of epinephrine. The edge-plane PG/SWCNTs-CoPc electrode displayed suitable electrocatalytic properties towards epinephrine oxidation with enhanced peak currents. At a pH of 7.4 in 0.1 M PBS, the fabricated sensor detected epinephrine with high sensitivity, low LOD, and good LOQ of 8.71 ± 0.31 A.M^−1^, 0.04 µM, and 1.31 µM, respectively. Good reproducibility, high stability, high selectivity, and sensitivity are the attributes of the electrode. It was also found that ascorbic acid did not interfere with epinephrine analysis. The sensor was employed to determine epinephrine in epinephrine tartaric acid injection solution with good recovery.

Yang and his research group [64] constructed an immunosensor using nanoCoPc-fMWCNTs/GCE to detect procalcitonin (PCT). Figure 3 illustrates the stepwise preparation of the nanoCoPc-fMWCNTs. The electrochemical study was performed via CV, while the PCT detection was performed in 0.1 M PBS at a pH of 7.4 using DPV. CoPc-MWCNTs/GCE produced electrochemical signals without needing additional redox mediators or labeling. Additionally, a pseudobienzyme system based on the catalytic properties of choline oxidase (ChOx) and CoPc was made to improve the sensor’s sensitivity. The authors found that the fabricated immunosensor demonstrated good performance for PCT with a low LOD of 1.23 pg mL^−1^ and a wide LDR of 0.01–100 ng mL^−1^. The fabricated immunosensor is a promising sensor for electrochemical detection of PCT in real biological samples (human serum).

The determination of nevirapine in the drug sample was performed by Kantize et al. [65] using a polymeric CoPc-nafion-CNTs composite modified on a platinum electrode. This was accomplished by consecutive drop-casting of a nanocomposite consisting of fMWCNTs and tetra-substituted coumarin CoPc (CoPc-cou), followed by immobilizing 5% Nafion perfluorinated resin solution (Naf-5). The CoPc-cou-f-MWCNTs/Naf-5/Pt electrode was used to determine antiretroviral drug nevirapine (NVP) in PBS (pH 12) via linear sweep voltammetry (LSV) and chronoamperometry. The LSV revealed an LOD and LDR of 0.2 nM and 0.6 nM to 30 µM, whereas the CA offered an LOD and LDR of 0.21 nM and 2.5 to 30 µM. This sensor demonstrated a better selectivity for NVP in the presence of interferents (uric acid, ascorbic acid, cysteine, dopamine, metronidazole) with very good recovery in spiked river water sample analysis.

In Table 3 below, the summary of the analytes detected with CoPc-MWCNT-modified electrodes with the techniques employed, LOD, and LDR is given.

### 2.4. Detection of Uric Acid, Glutathione, and Cysteine

Pari and Reddy [66] developed a simple and sensitive method for uric acid (UA) determination by modifying GCE with 2,4-dibromo-6-aniline-tetra (DBCMAT) substituted on CoPcMWCNTs via drop casting. Under optimum conditions, UA was determined in 0.1 M PBS at a pH of 7 via three techniques—CV, CA, and DPV. The DBCMAT-CoPc/MWCNTs/GCE-modified electrode demonstrated enhanced electrocatalytic activity and a lower potential for UA oxidation. The electrode delivered an LOD of 0.03 (CV), 0.066 (DPV), and 0.016 μM(CA) and an LDR of 0.1–1.8 (CV), 0.2–2.8 (DPV), and 0.05–0.8 μM(CA) with a sensitivity of 131.85 (CV), 22.634 (DPV), and 2.509 μAμM^−1^cm^−2^ (CA). The developed sensor demonstrated unique benefits, including low operating potential, good stability, high sensitivity, and remarkable repeatability and reproducibility for uric acid detection. In the presence of these molecules—cysteine, glucose, glycine, H_2_O_2_, and L-tyrosine—it was discovered that the sensor was highly selective for UA. The fabricated electrode was expediently applied for UA and DA determination in urine samples with good results. This electrode offers a lower LOD compared to the obtained result in [67].

Giarola and Pereira [67] further developed a voltammetric sensor to detect uric acid using CoPc-MWCNTs/GCE. Before this, the CoPc-MWCNTs composites were prepared by ultrasonic agitation of CoPc and MWCNTs in DMSO. The electrochemical behavior of UA at the CoPc-MWCNTs/GCE was investigated in 0.1 M PBS at a pH of 7 via CV. Under optimal conditions, the sensor delivered a wide LDR of 125–4000 μM with LOD and limit of quantitation (LOQ) of 260 and 860 μM, respectively, via the SWV technique. The electrode is characterized by good stability, sensitivity, and reproducibility. The presence of DA and AA do not interfere with UA detection, portraying good selectivity of the sensor. The fabricated sensor was successfully employed to determine UA in human urine samples.

Wang and his research group [55] investigated the fabrication of Pt-CoPc-MWCNTs/GCE to detect UA and DA simultaneously. The modified electrode was made by immobilizing GCE with MWCNTs covered with CoPc composite and platinum nanoparticles via in situ synthesis. The electrocatalyst displayed good electrochemical activity for uric acid (UA) and dopamine (DA). The LOD obtained for UA and DA were 1.4 and 2.6 μM, while the linear responses ranged from 5 to 100 μM and 5 to 170 μM, respectively. Notably, AA has no interference while simultaneously detecting UA and DA.

Gutierrez et al. [68] presented a GCE modified with pyrrole (Ppy), CoPc, and MWCNTs via electropolymerization to determine glutathione and cysteine. The mechanism oxidation of thiols in alkaline media as illustrated in Equations (6)–(9) involves the formation of a bond between the metal center in the phthalocyanine and the sulfur atom in the thiolate. The best electrocatalytic activity of the CoPc-MWCNTs-PPy/GCE electrode towards the analytes was obtained in 5 mM ferrocyanide solution in 0.5 M NaOH. The sensor delivered a sensitivity, LOD, and limit of quantification of 2.930 μA/mM, 0.03 mM, and 0.10 mM, respectively, for cysteine. For glutathione, the sensor offered 1.15 μA/mM, 0.02 mM, and 0.66 mM for sensitivity, LOD, and limit of quantification, respectively. Despite the low sensitivity of the electrode, it offered a low LOD and limit of quantification. Hence, they are good candidates as voltammetric sensors for biological samples.
RSH_sol_ + OH− ⇄ RS^−^_sol_ + H_2_O(6)
[M(II)Pc]_film_ + RS^−^_sol_ → [R-S–M(I)Pc]^−^_film_(7)
[R-S–M(I)Pc]^−^_film_ → [M(II)Pc]_film_ + RS^−^_sol_ + e^−^(8)
RS^−^_sol_ + RS^−^_sol_ → RS-RS_sol_(9)
where RSH represents thiol, MPc denotes metallophthalocyanine, and “film” and “sol” represent the film on the electrode and species in solution, respectively. Step 4 is rapid and irreversible. Step 2 is the most important, which involves Co-S bond formation, with a partial oxidation of the bound thiol molecule and a partial reduction of the metal center in the catalyst.

A similar electrochemical sensor for glutathione and l-cysteine detection was also fabricated by Argote et al. [69] by immobilizing CoPc and MWCNTs on GCE via electropolymerization of the pyrrole surfactant. The hybrid sensor displayed a good electrocatalytic behavior towards the oxidation of glutathione and l-cysteine in 0.1 M NaOH. The hybrid PyC_10_MIM^+^Br^−^-CoPc-MWCNT/GCE electrode offered a sensitivity of 5.240 and 6.733 μA/mM for glutathione and l-cysteine, respectively. The LOD and LOQ obtained for glutathione were 0.013 and 0.040 mM, respectively, while 0.014 and 0.043 mM were obtained for l-cysteine. The authors established that the pyrrole surfactant-derived hybrid electrodes have substantially smaller values, making them better suited for fabricating electrochemical thiols sensors in aqueous solutions. A low LOD obtained by this sensor is proof of its reliability for detecting thiols in biological samples. This hybrid sensor delivered a higher sensitivity with a better LOD than the results obtained in [68].

Sun and his research team [70] utilized the synergistic effect between CoPc and fMWCNTs to develop an aptasensor for kanamycin detection. The aptasensor was made by modifying a gold electrode (GE) surface with CoPc-MWCNTs nanocomposites. Under optimal conditions, in 0.1 M PBS at a pH of 7.4 via DPV, the CoPc-fMWCNTs/GE electrode displayed excellent stability, good reproducibility and repeatability, high sensitivity, high specificity, an LDR of 0.15–10 μM and a low detection limit (0.0058 μM). The fabricated sensor demonstrated a good selectivity for kanamycin amidst interferents such as chlortetracycline, chloromycetin, neomycin sulfate, and oxytetracycline. The aptasensor was successfully employed to detect kanamycin in the spiked milk sample.

A summary of the analytes detected with CoPc-MWCNTs modified electrodes with the techniques employed, LOD, and LDR is given in Table 4 below.

### 2.5. Detection of lactic Acid, Glucose, and Hydrazine

In an attempt to detect lactic acid, Shao and group [19] fabricated a novel sensor by modifying GCE with cobalt polyphthalocyanine (CoPPc) nanocomposite and MWCNTs-COOH. The reduction mechanism of lactic acid on the CoPPc/MWCNTs-COOH/GCE surface is illustrated in Figure 6 below. The electrocatalytic activities of the CoPPc/*f*MWCNTs/GCE towards lactic acid detection was assessed via CV in 0.1 M PBS at a pH of 4. The fabricated electrode exhibited outstanding electrochemical performance for lactic acid reduction over a wide LDR of 10–240 μM and a low LOD of 2 μM. It also displayed a high selectivity against common interfering molecules (glucose, ascorbic acid, sodium chloride, dopamine, hydrogen peroxide, and uric acid). Notably, the sensor was effectively used for lactic acid determination in rice wine samples, demonstrating the excellent potential for quick monitoring applications.

Porto and his group [72] developed a sensor to determine pyridoxine by modifying a pyrolytic graphite electrode (PGE) with CoPcMWCNTs composite. The mechanism of the pyridoxine oxidation on the surface of the CoPcMWCNT/PGE is illustrated in Figure 7 below. Under optimum conditions of 0.3 M PBS at a pH of 5.5, the CoPcMWCNT/PGE was employed to detect pyridoxine (vitamin B6) via the DPV technique. The LOD, LOQ, and LDR obtained were 0.50 μM, 1.67 μM, and 10–400 μM, respectively. It also offers the benefits of a quick response, low cost, and a sensitivity of 0.037 μAL μmol^−1^, testifying to the technique’s excellent sensitivity. The sensor was conveniently applied for pyridoxine determination in real samples of pharmaceutical formulations (RSD < 5%), indicating the suitability of the developed electrode for accurate pyridoxine detection in pharmaceutical formulations containing pyridoxine.

Mounesh and his research team [73] developed an amperometric sensor for glucose by modifying tetracinnamide cobalt phthalocyanine (TCIDCoPc) and MWCNTs on GCE. The modified TCIDCoPc-MWCNTs/GCE exhibited excellent electrocatalytic properties and reduced potential for glucose oxidation. Under optimized conditions, at a pH of 7 in 0.1 M PBS, the developed sensor gave a wide LDR from 2 to 20 (CV); 2 to 12 (DPV); 5 to 50 (CA) mM/L, detection limit of 0.9 (CV); 5.33 (DPV); 6 (CA) mM, and sensitivity of 1.905 (CV); 3.483 (DPV); and 1.035 (CA) mA mM^−1^ cm^−2^. The fabricated sensor was characterized by quick response time, good sensitivity, reduced working potential, and good reproducibility and repeatability. The developed sensor also demonstrated a better selectivity for glucose amidst interfering biological species such as UA, DA, and AA.

In another study conducted by Devasenathipathy and his team [74], an amperometric biosensor based on MWCNT cobalt tetrasulfonated phthalocyanine (CoTsPc-MWCNT) nanocomposite-modified GCE was fabricated for glucose detection. The fabricated electrode tagged MWCNT–CoTsPc/GCE displayed an excellent electrocatalytic activity towards detecting glucose in 0.1 M NaOH using an amperometry technique. A reasonable sensitivity (122.5 µA mM^−1^ cm^−2^), a wide LDR (10 µM–6.34 mM), along with an LOD of 0.14 µM were reported for this sensor. The proposed sensor is characterized by high stability, good sensitivity, low working potential, repeatability, reproducibility, and quick response time (2 s). Amperometry study established that the fabricated sensor has high selectivity for detecting glucose amidst interfering molecules such as AA, DA, UA, galactose, fructose, lactose, and sucrose. The sensor was practically employed to detect glucose in serum samples from human blood.

Mounesh and Reddy [58] developed another amperometric sensor for hydrogen peroxide (H_2_O_2_) and glucose by modifying GCE with synthesized tetra-cobalt(II) carboxamide-PEG_2_-biotin phthalocyanine composite (CoTPEG_2_BAPc) and MWCNTs via drop-casting. Under optimized conditions of pH 7 in 0.1 M PBS, the developed biosensor delivered a wide LDR of (CA: 5–50, DPV: 2–22; CV: 5–25 µmol) and (CA: 5–50, DPV: 2–22; CV: 2–16 µmol) for H_2_O_2_ and glucose, respectively, LOD of (CA: 12.5; DPV: 2; CV: 0.33 µM) and (CV: 1.5; DPV: 5; CA: 10 µM) for glucose and H_2_O_2_, and a good sensitivity of (CA: 0.101; DPV: 1.978; CV: 0.947 µA µM^−1^ cm^−2^) and (CA: 0.162; DPV: 1.888; CV: 1.250 µA µM^−1^ cm^−2^) for glucose and H_2_O_2_. The biosensor is characterized by a rapid response time, good sensitivity, repeatability, and reproducibility. It also demonstrated a high selectivity for glucose and H_2_O_2_ amidst interfering molecules such as L-cysteine, glycine, DA, UA, and AA. The remarkable selectivity of the fabricated sensor is attributed to the comparatively low potential for detection, which greatly minimizes the responses of typical electroactive interference. Likewise, the addition of MWCNTs to the modified electrode also helped to reduce interference since their neutral charge prevented interfering molecules from penetrating the electrode’s surface. The practicability of the electrode was demonstrated by the successful detection of H_2_O_2_ and glucose in human blood samples and contact lens care solution, respectively.

Again, Mounesh et al. [71] fabricated a sensor to detect hydrazine and L-cysteine by modification of GCE with a poly (L-lactide)-carboxamide-cobalt phthalocyanine composite (TPLLCA-CoPc) and MWCNTs. Under optimized conditions of pH 7 in 0.1 M PBS, the modified TPLLCA-CoPc/MWCNTs/GCE detected L-cysteine and hydrazine in the nanomolar. The modified electrode offered a linear range of (CA: 5–50, DPV: 1–10, CV: 2–10 nmol L^−1^) for L-cysteine and hydrazine, a low LOD of (CA: 3.33, DPV: 1, CV: 1.33 nmol) and (CA: 6, DPV: 0.033, CV: 0.33 nmol) for L-cysteine and hydrazine, respectively, and high sensitivity of (CA: 0.945, DPV: 4.325, CV: 1.299 μA nM^−1^ cm^−2^) and (CA: 0.770, DPV: 4.193, CV: 1.719 μA nM^−1^ cm^−^^2^) for hydrazine and L-cysteine, respectively. The TPLLCA-CoPc/MWCNTs/GCE exhibits high stability and sensitivity, good reproducibility, and repeatability. The authors reported that these molecules (dopamine, glucose, ascorbic acid, glycine, uric acid, and hydrogen peroxide) do not interfere with the detection of L-cysteine and hydrazine.

Table 5 below summarizes the analytes detected with CoPc-MWCNT-modified electrodes with the techniques employed, LOD, and LDR.

Aside from these, CoPcMWCNTs nanocomposite have been utilized in various other processes such as oxidation of styrene to benzaldehyde [7,75,76], catalytic oxidation of benzyl alcohol to benzaldehyde [6,25], electrochemical reduction of oxygen [77,78,79,80,81], electrochemical conversion of CO_2_ to CO [82], removal of mercaptan from natural gas [83], photocatalytic oxidation of butan-2-ol [84], a catalyst for microbial fuel cell [85], catalyst in glucose/O_2_ fuel cells [86], and electrochemical reduction of carbon dioxide and carbon monoxide to methanol [87]. The conversion rate and selectivity recorded in these processes were satisfactory.

The use of cobalt phthalocyanine (CoPc) has been favored over other MPcs owing to its greater advantages such as high charge transfer capabilities [88], stability, high catalytic current density [89], reduced overpotential, diversify preparation methods, and high water-solubility [41,42,43]. It is also important to note the challenges researchers [75,90] encountered while using CoPc-MWCNTs nanocomposite as a sensing material. CoPc suffers from aggregation and hydrophobicity due to huge π-π-conjugated systems. This involves the aggregation of MPc inside the reaction medium to form either a polymer or an inactive dimer due to the structural characteristics of the metal Pc itself. The dimers’ formation reduces the axial ligand’s active point; meanwhile, the majority of the catalytic process takes place at the axial position, thereby leading to a significant reduction in catalytic activity [7]. A single dispersed molecule in the reaction medium was employed to overcome this setback and improve the catalytic activity of the MPc. The MPc aggregation between molecules can be effectively prevented by immobilizing it on a solid carrier [90]. Silica, carbon nanotubes, activated carbon, zeolites, and graphene are the most used carriers. The addition of the carrier enables the Pc molecules to be evenly distributed on its surface, exposing more active sites that can increase the contact possibility of the catalyst with the target substrate and speed up the reaction [7]. Alternatively, Pc aggregation can be prevented by uniformly loading Pc onto MWCNTs using an ultrasonic approach. As a result, the agglomeration impact of Pcs is significantly diminished, allowing for an increase in the catalytic surface area and an improvement in catalytic efficiency [75].

Going by Table 1, Table 2, Table 3, Table 4 and Table 5, the CoPc-MWCNT-nanocomposite-modified electrodes have been successfully employed to detect a large variety of analytes such as ascorbic acid, acetaminophen, carbaryl, cysteine, epinephrine, diethylstilbestrol, dopamine, glucose, glutathione, hydrazine, hydrogen peroxide_,_ kanamycin, nevirapine, nitrite, lactic acid, paracetamol, procalcitonin, pyridoxine, uric acid, Cd^2+^, and Pb^2+^. These CoPc-MWCNT-nanocomposite-modified electrodes achieved a very low LOD, wide LDR, high sensitivity, and selectivity required for analyzing trace amounts of these analytes. The lowest LOD (0.033 nM) detected by this modified electrode was obtained from the detection of hydrazine using the DPV technique, followed by nevirapine (0.2 nM) using the CA technique.

## 3. Conclusions

In this review, we extensively summarized the efforts made in fabricating electrochemical sensors using CoPc-MWCNTs nanocomposite along with the various analytes detected by this sensing material. This nanocomposite was modified with various electrodes such as GCE, gold electrode, screen-printed electrode, platinum electrode, pyrolytic graphite electrode, and paraffin composite electrode, as shown in Table 1, with GCE being mostly used. Emphasis was also made on the different fabrication methods and the detection techniques used with their supporting electrolytes/pH, LOD, and LDR. Owing to these extraordinary properties of CoPc-MWCNTs nanocomposite, more studies are expected in the future to determine other analytes that this sensing material has not determined.

## Data Availability

Data are available upon request from authors.

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
