# Peer review of "Recent Advances in the Use of CoPc-MWCNTs Nanocomposites as Electrochemical Sensing Materials"

_biosensors, 2022, doi:10.3390/bios12100850_

Round 1

Reviewer 1 Report

In this mini review, authors presented the introduction of CoPc and MWCNTs and the composite CoPc-MWCNTs applications in sensing fields. However, lack of comment on the sensing mechanism resulted in less novel meaningfulness.

1.      Discussion on effect of CoPc-MWCNTs made by different preparation methods on the sensing performance is missing.

2.      The advantage of CoPc-MWCNTs should be highlighted in comparison way with other metal ions centered Pc species.

3.      The tables regarding the analytical merit of figure should be separately given in each portion of target analyst for clear comparison.

4.      For a good review, the merits, and drawbacks for the CoPc-MWCNTs-based sensing assays should be commented in detail and the abstract-style writing on the sensing applications should be corrected.

5.      What are the main principles for the sensing selectivity of CoPc-MWCNTs to detect various target samples?

6.      Additionally, the statement of CNTs that was much better than that of other carbon electrodes is not real story.

7.      The abbreviation in the first place in main context should be followed by full name.

8.      English writing should be polished. For instance, it is a poor sentence that the applicability of the fabricated electrode for accurate UA and DA detection was carried out using urine samples with good results. Here, a lower LOD was obtained compared to the obtained result in [36].

Author Response

The Editor

Biosensors

REVISION OF MANUSCRIPT SUBMITTED FOR PUBLICATION

Manuscript Title:  Recent Advances in the Use of CoPc-MWCNTs Nanocomposites as Electrochemical Sensing Materials

Manuscript ID: nanomaterials-1941241

We appreciate the reports on our manuscript. As a result of this, we submit a response to the comments on the manuscript for further consideration.

The reviewers' efforts at improving this manuscript are well appreciated. We have carefully considered the comments. The responses to all reviewers' comments are highlighted in yellow in the manuscript. 

Response to the Reviewer’s Comments

Reviewer 1 Comments

  1. Comment: However, lack of comment on the sensing mechanism resulted in less

novel meaningfulness.

Response: The mechanism of various processes has been included except those

unavailable in the source reference. See page 5 lines 159 – 174, See page 5 lines 181 – 189, See page 5 & 6 lines 198 – 204, See page 7 lines 256 – 261, See page 8 lines 313 – 318, See page 10 lines 351 – 352 & 359, See page 14 lines 485 – 488 & 497 – 506, and others 

  1. Comment: Discussion on effect of CoPc-MWCNTs made by different preparation

           Methods on the sensing performance is missing.

Response:  The different methods of preparation of CoPc-MWCNTs have been included. See page 4 lines 149 – 154, See page 7 & 8 lines 240 – 243, See page 10 lines 343 – 351, See page 11 lines 389 – 392, and others 

  1. Comment: The advantage of CoPc-MWCNTs should be highlighted in comparison

           with other metal ions centered Pc species.

Response: The suggestion has been included. See page 19 lines 667 – 670.

  1. Comment: The tables regarding the analytical merit of figure should be separately

           given in each portion of target analyst for clear comparison.

Response: The table has been restructured to accommodate the suggestion. See table 1 – 5 (Pages 6, 8, 13, 15 & 18)

  1. Comment: For a good review, the merits, and drawbacks for the CoPc-MWCNTs-

based sensing assays should be commented in detail and the abstract-style writing on the sensing applications should be corrected.

Response: The suggestion has been included. See page 19 lines 667 – 687.

  1. Comment: What are the main principles for the sensing selectivity of CoPc-MWCNTs to detect various target samples?

Response: The sensing selectivity of CoPc-MWCNTs have been included. See page 7 lines 248 – 251, See page 17 lines 617 – 621.

  1. Comment: Additionally, the statement of CNTs that was much better than that of

other carbon electrodes is not real story.

Response: The statement has been removed.

  1. Comment: The abbreviation in the first place in main context should be followed by full name.

Response: Correction made; see page 1. lines 13 & 34 – 35, page 2, lines 41 – 42, page 3, line 111, page 4, lines 141 – 142, page 6, lines 207 – 208, and other similar

places.

8. Comment: English writing should be polished. For instance, it is a poor sentence that the applicability of the fabricated electrode for accurate UA and DA detection was carried out using urine samples with good results. Here, a lower LOD was obtained compared to the obtained result in [36].

Response: The sentence has been restructured; see page 13, lines 465 – 467.

Reviewer 2 Report

The review manuscript “Recent Advances in the Use of CoPc-MWCNTs Nanocomposites as Electrochemical Sensing Materials” is a very interesting subject for the scientific community of (bio)sensors, as it presents two materials that are very promising for several sub-areas of electroanalysis. I have some comments that can enhance the quality of review.

-        There are some important characteristics of CoPc thar must be added in the introduction section. One of them is the easy replacement of the organic groups in the axial and equatorial positions of the complex, which gives several functionalities to anchoring these complexes in solid substrates. Some actual references have been explored this issue : 10.1007/s12274-021-3962-2; 10.1007/s00604-022-05360-z; 10.1016/j.solidstatesciences.2022.106905.

-        The water-soluble CoTsPc is applied in several sensors because it’s capacity of film forming together of a polyelectrolyte positively charged, giving sensors using LbL method and LB method of film forming. It should be investigated.

-        MWCNTs are inevitable materials for sensors. The capacity of enhance current of several analytes is very promising. A recent literature of CNTs are preferable instead some non-actual.

-        The authors must explore the interactions that were observed in the formation of sensors with CoPc and CNTs (i.e. -COOH of MWCNT with -NH2 terminal of CoPc)... The formation of composite is invited to be described.

Author Response

The Editor

Biosensors

REVISION OF MANUSCRIPT SUBMITTED FOR PUBLICATION

Manuscript Title:  Recent Advances in the Use of CoPc-MWCNTs Nanocomposites as Electrochemical Sensing Materials

Manuscript ID: nanomaterials-1941241

We appreciate the reports on our manuscript. As a result of this, we submit a response to the comments on the manuscript for further consideration.

The reviewers' efforts at improving this manuscript are well appreciated. We have carefully considered the comments. The responses to all reviewers' comments are highlighted in yellow in the manuscript. 

Reviewer 2 Comments

  1. Comments: There are some important characteristics of CoPc that must be added in the introduction section. One of them is the easy replacement of the organic groups in the axial and equatorial positions of the complex, which gives several functionalities to anchoring these complexes in solid substrates. Some actual references have been explored this issue: 10.1007/s12274-021-3962-2; 10.1007/s00604-022-05360-z; 10.1016/j.solidstatesciences.2022.106905.

Response: The suggestion has been included in the introduction section. See page 2 lines 70 – 79.

  1. Comments: The water-soluble CoTsPc is applied in several sensors because it’s capacity of film forming together of a polyelectrolyte positively charged, giving sensors using LbL method and LB method of film forming. It should be investigated.

Response: The suggestion has been investigated and included in the introduction section. See page 3 lines 89 – 92.

  1. Comments: MWCNTs are inevitable materials for sensors. The capacity of enhance current of several analytes is very promising. A recent literature of CNTs is preferable instead some non-actual.

Response: The paragraph has been restructured to accommodate the suggestion. See page 2 lines 42 – 47.

  1. Comments: The authors must explore the interactions that were observed in the formation of sensors with CoPc and CNTs (i.e. -COOH of MWCNT with -NH2 terminal of CoPc). The formation of composite is invited to be described.

Response: suggestion on CoPcMWCNTs formation has been included. See page 3 lines

80– 86.

Round 2

Reviewer 1 Report

The revision could be accepted for publication as it is.

Reviewer 2 Report

The review has been enhanced the quality and therefore deserves be published in Biosensors.